# Augmented Reality Surgical Navigation System Integrated with Deep Learning

**DOI:** 10.3390/bioengineering10050617

**Published:** 2023-05-20

**Authors:** Shin-Yan Chiou, Li-Sheng Liu, Chia-Wei Lee, Dong-Hyun Kim, Mohammed A. Al-masni, Hao-Li Liu, Kuo-Chen Wei, Jiun-Lin Yan, Pin-Yuan Chen

**Affiliations:** 1Department of Electrical Engineering, College of Engineering, Chang Gung University, Kwei-Shan, Taoyuan 333, Taiwan; 2Department of Nuclear Medicine, Linkou Chang Gung Memorial Hospital, Taoyuan 333, Taiwan; 3Department of Neurosurgery, Keelung Chang Gung Memorial Hospital, Keelung 204, Taiwan; 4Department of Electrical and Electronic Engineering, College of Engineering, Yonsei University, Seodaemun-gu, Seoul 03722, Republic of Korea; 5Department of Artificial Intelligence, College of Software & Convergence Technology, Daeyang AI Center, Sejong University, Seoul 05006, Republic of Korea; 6Department of Electrical Engineering, National Taiwan University, Taipei 106, Taiwan; 7New Taipei City Tucheng Hospital, Tao-Yuan, Tucheng, New Taipei City 236, Taiwan

**Keywords:** surgical navigation, mixed reality, augmented reality, neurosurgery, deep learning, automatic scanning, EVD surgery

## Abstract

Most current surgical navigation methods rely on optical navigators with images displayed on an external screen. However, minimizing distractions during surgery is critical and the spatial information displayed in this arrangement is non-intuitive. Previous studies have proposed combining optical navigation systems with augmented reality (AR) to provide surgeons with intuitive imaging during surgery, through the use of planar and three-dimensional imagery. However, these studies have mainly focused on visual aids and have paid relatively little attention to real surgical guidance aids. Moreover, the use of augmented reality reduces system stability and accuracy, and optical navigation systems are costly. Therefore, this paper proposed an augmented reality surgical navigation system based on image positioning that achieves the desired system advantages with low cost, high stability, and high accuracy. This system also provides intuitive guidance for the surgical target point, entry point, and trajectory. Once the surgeon uses the navigation stick to indicate the position of the surgical entry point, the connection between the surgical target and the surgical entry point is immediately displayed on the AR device (tablet or HoloLens glasses), and a dynamic auxiliary line is shown to assist with incision angle and depth. Clinical trials were conducted for EVD (extra-ventricular drainage) surgery, and surgeons confirmed the system’s overall benefit. A “virtual object automatic scanning” method is proposed to achieve a high accuracy of 1 ± 0.1 mm for the AR-based system. Furthermore, a deep learning-based U-Net segmentation network is incorporated to enable automatic identification of the hydrocephalus location by the system. The system achieves improved recognition accuracy, sensitivity, and specificity of 99.93%, 93.85%, and 95.73%, respectively, representing a significant improvement from previous studies.

## 1. Introduction

In recent years, several studies have proposed applying augmented reality (AR), virtual reality (VR), and artificial intelligence (AI) technologies with medicine, producing promising results but also exhibiting some limitations. For instance, AR or VR technology can be a promising tool for complex procedures, especially in maxillofacial surgery, to ensure predictable and safe outcomes [1]. AR-based surgical navigation techniques can be broadly categorized into three types: projection navigation [2], optical positioning navigation [3,4,5,6], and image positioning navigation [7,8,9,10]. Projection navigation involves direct projection onto the surgical area [9] or a semi-transparent mirror for superimposition [11].

In terms of projection navigation, Tabrizi et al. [2] proposed a direct projection navigation approach using a projector to project onto the Region of Interest (ROI) for brain tumor removal. However, this approach causes curvature distortion and reduces accuracy. Fida et al. [12] applied volume subtraction navigation to knee joint transplantation (KNS), projecting the image onto a semi-transparent mirror for superimposition, with two optical cameras to achieve better depth detection and put the real-time image into the tracking learning detection (TLD) method [13] to track specific objects. The system processes the frames using volume subtraction to remove redundant images and improve the accuracy of KNS surgery. Nonetheless, due to space limitations, volume subtraction navigation is not suitable for neurosurgical procedures such as extra-ventricular drainage (EVD). Regarding optical positioning navigation, a positioning device (NDI Polaris Vicra and a cursor ball), along with a screen (a head-mounted device (HMD)), were utilized by Chen et al. [3]. The system determines the current virtual image’s position based on the distance between the positioning devices and the cursor ball, and subsequently applies iterative closest point (ICP) and manual adjustment to superimpose the virtual image onto the physical image.

Concerning image positioning navigation, Müller et al. [5] suggested fiducial marker image navigation, which employs a lens to recognize fiducial points and generate virtual images for percutaneous nephrolithotomy (PCNL). Nevertheless, the reference point placement before surgery is time-consuming (99 s), and the average error of 2.5 mm is unsuitable for neurosurgical procedures (such as EVD).

According to Prakosa et al. [14], the use of AR guidance in a virtual heart can increase catheter navigation accuracy, leading to a potential effect on ventricular tachycardia (VT) termination. In their study, the use of AR resulted in significantly smaller errors in 54 VT 1 ablation simulations compared to those without AR. The accuracy of VT termination may also improve with substrate characteristics. Tu et al. [15] proposed a HoloLens-to-world registration method that utilizes an external EM tracker and a customized registration cube to improve depth perception and reduce registration errors. The preclinical phantom experiments showed reprojection errors along the X, Y, and Z axes to be 1.55 ± 0.27 mm, 1.71 ± 0.40 mm, and 2.84 ± 0.78 mm, respectively. The end-to-end evaluation method indicated the distance error was 1.61 ± 0.44 mm, and the 3D angle error was 1.46 ± 0.46°. However, the accuracy performances of these methods are still not satisfactory, making them unsuitable for current surgical operations.

Over the past decade, there has been significant attention given to advances in deep learning methods for various medical imaging applications such as diagnosis, segmentation, and detection [16,17,18,19]. Convolutional neural networks (CNNs) have been widely used in deep learning for image-based prediction [20,21,22]. Developers have also incorporated decoder paths into the CNN to segment objects within input images, resulting in the generation of segmentation maps with pixel-wise predictions that match the input size. This approach, known as encoder-decoder networks or U-Net [23], is commonly employed in most medical imaging applications that use supervised learning, which requires labeled references during the training phase.

Long et al. [24] proposed a fully convolutional network that takes the input of the arbitrary size and produces correspondingly-sized output with efficient inference and learning. They adapted several classification networks (AlexNet [20], the VGG net [21], and GoogLeNet [22]) into fully convolutional networks and transferred their learned representations by fine-tuning [25] to the segmentation task.

Yigin et al. [26] explored the effectiveness of commonly used morphological parameters in hydrocephalus diagnosis. For this purpose, they compared the effect of six common morphometric parameters (Frontal Horns’ Length (FHL), Maximum Lateral Length (MLL), Biparietal Diameter (BPD), Evans’ Ratio (ER), Cella Media Ratio (CMR), and Frontal Horns’ Ratio (FHR)) in terms of their importance in predicting hydrocephalus using a Random Forest classifier.

Martin et al. [27] proposed a method using Compositional Pattern Producing Network (CPPN) to enable Fully Convolutional Networks (FCN) to learn cerebral ventricular system (CVS) location. To address the ventriculomegaly problem that arises in the clinical routine, dilation of the CVS is required.

However, current AI methods cannot automatically locate hydrocephalus. To address this issue, this paper proposes a comprehensive solution to obtain the surgical target, scalpel entry point, and scalpel direction, and automatically locate hydrocephalus. The proposed approach includes virtual object automatic scanning operation navigation to improve accuracy and the use of a tablet computer lens to align two custom image targets for locating the virtual head and virtual scalpel. The improved U-Net [28] is also utilized to recommend target points, resulting in a surgery efficiency and accuracy rate of 99.93%. Clinical trials were conducted for EVD (extra-ventricular drainage) surgery, and surgeons have confirmed the feasibility of the system. Overall, the proposed approach has the potential to enhance surgical outcomes and improve patient care in the field of neurosurgery.

## 2. Materials and Methods

### 2.1. System Overview

The proposed system comprises an AR device (Surface Pro 7 (Microsoft Co., Redmond, WA, USA)), printed images for head and scalpel positioning, an upright tripod, a flat clamp, and a medical septum cover. The tablet tracks the feature points of the two positioning images to display the virtual image correctly (see Figure 1a). The system follows the flow chart depicted in Figure 1b, providing multiple functions such as automatic scanning of DICOM-formatted virtual objects, selection of surgical targets, generation of surgical trajectories, and color-assisted calibration of surgical entry points to aid surgeons during EVD surgery. Table 1 defines the symbols and parameters used in the proposed method, while Table 2 presents the main abbreviations employed in this article.

### 2.2. Virtual Object Automatic Scanning

The patient’s head CT scans are converted into a virtual head object with Avizo software (Waltham, MA, USA) [29], which provides an image analysis platform for identifying the locations of the scalp, skull, and hydrocephalus. The principle of the automatic scanning method is based on a trigger function. Virtual planes that are perpendicular to the *x*, *y*, and *z* axes in Unity are utilized to scan virtual objects. The entry and exit points (or two collision points) are obtained when these virtual planes enter and leave the virtual object or when they enter the object from two opposite directions, resulting in a total of six reference points.

As an example, RP_x_^R^ (reference point on the right of the *X*-axis) is obtained when the axial plane enters the virtual head object from the right, while RP_x_^L^ (reference point on the left of the *X*-axis) is obtained when the axial plane enters the virtual scalp from the left. RP_x_^R^ and RP_x_ serve as two reference points on the *X*-axis for displaying real-time DICOM (Digital Imaging and Communications in Medicine) images.

The system performs simultaneous virtual scalp scans along the three-axis (sagittal, frontal, horizontal, and lateral) to obtain a total of six reference points, which are displayed as virtual head objects as RP_x_^R^, RP_x_^L^, RP_y_^T^, RP_y_^B^, RP_z_^T^, and RP_z_^B^. These reference points play a critical role in locating the DICOM plane, which significantly impacts target accuracy (refer to Section 2.5 for detailed methodology). After completing the calibration, an accurate augmented reality environment is generated to assist with EVD surgical navigation.

### 2.3. Using Machine Learning to Predict and Recommend Surgical Target

To predict the location of the hydrocephalus and identify the surgical target, the system selected the connection area with the largest hydrocephalus, computed the center point of this area, and marked it as the final recommended surgical target point. The system’s connection areas were selected based on the segmented output of U-Net, with the largest connected pixels chosen using the bwconncomp function in Matlab. This function is designed to identify and count connected components in binary segmented images.

DICOM files were utilized to conduct cross-validation on ten groups of patients, with the database divided into five folds. Each group was assigned a number from 1 to 10, and two groups of patient DICOM data were used as testing sets, while the remaining groups served as training sets. Consequently, there were five folds with five distinct testing sets, with four folds (consisting of eight groups of patients’ DICOM data) as the training set and the other fold (the DICOM data of the other two groups) as the test set. The training set included label data and normalized data from the eight patient groups, while the test set used the label and normalized data from the other two groups. The normalized data was used as the basis for the final accuracy test.

### 2.4. Manual Operation Target Point Positioning

After superimposing the virtual head onto the real head, DICOM images in the horizontal and lateral planes are displayed on the tablet. This allows the surgeon to individually select and confirm the accuracy of the target position. Once the target position is confirmed, the relative position is then converted into conversion space, and the specific DICOM slice containing the ideal target ($N_{target}$-th slice) can be obtained from the total number of DICOM slices $N_{total}$. The DICOM image is displayed in the lower left of the screen, with the ideal target position displayed as ${Pos}_{target}^{2D}\left(X_{target}^{2D},Y_{target}^{2D}\right)$. The length and width of the DICOM image are $I_{dicomX}^{2D}$ and $I_{dicomY}^{2D}$, respectively. The origin point ${Pos}_{o}^{3D}\left(0,0,0\right)$ is located in the lower left of the head, with the length of the head along the X, Y, and Z axes being $I_{X}^{3D}$, $I_{Y}^{3D},I_{Z}^{3D}$, respectively. The target position in space is denoted by
(1)$${Pos}_{target}^{3D}\left(X_{target}^{2D}\times \frac{I_{X}^{3D}}{I_{dicomX}^{2D}},\frac{I_{Y}^{3D}}{N_{total}}\times \left(N_{target}-1\right),Y_{target}^{2D}\times \frac{I_{Z}^{3D}}{I_{dicomY}^{2D}}\right)$$

### 2.5. DICOM Image Real-Time Display and Selection of Target Point

After automatic scanning and obtaining the six reference points on the three axes, the longest distance of the reference point on each axis is calculated as *Dis_x_* (the distance on the *x*-axis), *Dis_y_* (the distance on the *y*-axis), and *Dis_z_* (the distance on the *z*-axis). The resulting values are then divided by the number of DICOM slices on the corresponding axis, including Num_x_, Num_y_, and Num_z_. The resulting values are the thicknesses on the *x*, *y*, and *z* axes, denoted as *T_x_*, *T_y_*, and *T_z_*, respectively.
(2)$$T_{x}=Dis_{x}÷Num_{x}$$
(3)$$T_{y}=Dis_{y}÷Num_{y}$$
(4)$$T_{z}=Dis_{z}÷Num_{z}$$

The distance between the scalpel’s edge point ($E_{p}$) and ${RP}_{x}^{L}$ is divided by $T_{x}$ to determine the corresponding DICOM slice on the *x*-axis, known as TrueX. This algorithm is repeated for the *y* and *z* axes.
(5)$$TrueX=\left(E_{p}-{RP}_{x}^{L}\right)÷T_{x},E_{p}\geq RP_{x^{L}}$$
(6)$$TrueY=\left(E_{p}-{RP}_{y}^{L}\right)÷T_{y},E_{p}\geq RP_{y^{L}}$$
(7)$$TrueZ=(E_{p}-{RP}_{z}^{L})÷T_{z},E_{p}\geq RP_{z^{L}}$$

Once TrueX and TrueZ have been calculated, the Unity Web Request function is utilized to present a real-time DICOM image display (Figure 2a) in combination with augmented reality. This allows surgeons to access the display without having to look away to a separate screen. Surgeons can then simply tap on the screen to select the ideal target (Figure 2b).

### 2.6. Generation of Surgical Trajectory and Calibration of Entry Point Angle

Once the target point is selected, a surgical simulation trajectory is generated, connecting the scalpel’s edge to the target point. The surgeon then confirms this trajectory by pressing the function button, which generates the final trajectory connecting the surgical entry point to the target point (Figure 3a). To ensure maximum accuracy of the location and path of the surgical entry point, color-assisted angle calibration is used. The color of the trajectory changes from red (Figure 3b) to yellow (Figure 3c) to green (Figure 3d), providing high color accuracy during surgery.

## 3. Experiments Studies and Tests

To demonstrate the feasibility of the method proposed in Section 2, prosthesis experiments were first conducted in the laboratory using the proposed method. Subsequently, clinical feasibility tests are carried out in hospitals. A Surface 7 tablet was used as the AR device in both test reports. Furthermore, Hololens 2 smart glasses are currently the most popular advanced medical AR HMD devices. A detailed explanation of the potential outcomes when substituting the AR devices with the Hololens 2 smart glasses is provided.

### 3.1. Experiment of the Virtual Object Automatic Scanning

To evaluate the accuracy of the proposed virtual object automatic scanning method, DICOM data from ten patients were utilized. The automatic scanning error was determined by measuring the distance between the predicted axis plane (automatically selected by the deep learning system) and the actual axis plane (where the target point was located). The minimum error, maximum error, lower quartile, median, and upper quartile, as well as the target point error established by the software Avizo, were also calculated. The virtual model of the point was imported into 3ds Max (Autodesk Inc., San Francisco, CA, USA) for anchor point processing, ensuring constant relative positions of the scalpel and the target point, which facilitated error calculation.

Additionally, it should be noted that the predicted axis plane was obtained by selecting the center point of the largest connection area as the target point after predicting the contour of the ventricle using the deep learning model. The actual axis plane was obtained by extracting the center point of the 3D object of the target point, which is created in 3ds Max from the DICOM data. Before this, the 3D object of the ventricle was generated in 3ds Max, followed by the 3D object of the target point that corresponds to the doctor-marked target point on the DICOM.

### 3.2. Experiment of Machine Learning to Predict and Recommend Surgical Target

To predict the location of hydrocephalus, U-Net (the original model) was employed for deep learning to maximize accuracy and minimize loss by setting 30 epochs, and identifying all ventricular regions in the patient’s DICOM images (Figure 4b). The Labeled ventricle contour (green) and the predicted ventricle contour (red) were drawn using Matlab (MathWorks Inc., Natick, MA, USA) (Figure 4a). Finally, the average sensitivity, specificity, and accuracy for predicting the location of hydrocephalus were calculated, and these data (sensitivity, specificity, and accuracy) are the result of comparing the contour with the pixel-by-pixel method.

The U-Net (the original model) architecture consists of two paths, the encoder, and the decoder. The encoder path, comprising convolutional and pooling layers, is responsible for extracting and learning contextual features. Conversely, the decoder path, which includes transpose convolution and up-sampling layers, aims to transfer these learned features into a single prediction layer of the same size as the input image, known as the dense prediction or segmentation map. Each convolution layer is appended with batch normalization and ReLU activation function, except for the last layer, which produces a binary output using sigmoid activation. The entire network uses a convolution kernel of size 3 × 3 and stride of 1 with feature maps of 32, 64, 128, 256, and 320 across all resolution levels.

Hyper-parameters used in this study were a learning rate of 0.003, a batch size of 20, and a total of 30 epochs. The objective was to minimize the overall loss of each pixel by computing the dice loss function between the segmented map and labeled reference, and the Adam optimizer was utilized to optimize the weight parameters in each layer.

### 3.3. Test of Clinical Feasibility

The feasibility of the proposed system was tested at various stages of clinical implementation, beginning with the conversion of DICOM images from patients into 3D models. To assess the clinical feasibility of each step, a pre-operative simulation was conducted in the operating room approximately 1 h before the surgery. Figure 5 illustrates the setup of the system in the operating room and its operation by a neurosurgeon. Specifically, Figure 5a shows the superimposed 3D model of the patient’s head position, while Figure 5b shows the DICOM-selected target position on the display. Figure 5c depicts the alignment position and angle following entry point selection, and Figure 5d shows the completed alignment. Following the entire process, an experienced surgeon concluded that the system concept is feasible for clinical use.

### 3.4. Test of Hololens 2 Feasibility

In order to test whether our proposed method is accurate on HoloLens 2, we designed a separate accuracy experiment specifically for HoloLens 2. A solid sponge brick was used for flower arrangement, and the target point was set as the middle point at the bottom of the sponge brick. A virtual sponge brick model of the same size was created in Unity, and the target point, insertion point, and guide path were set. The experimenters wore Hololens 2 and inserted the real navigation stick into the sponge brick through the virtual guide path seen in the Hololens 2 screen to test whether it could accurately reach the target point.

To perform the experiment, the experimenter needed to superimpose the real sponge brick and the virtual model (Figure 6), align the navigation stick with the path of the virtual guide (Figure 6b), insert the sponge brick along the guiding direction, and pass the navigation stick through the sponge bricks. The difference between the “true target position” and “the final position where the experimenter arrived at the real sponge brick using the navigation stick” was measured to calculate the error distance.

## 4. Results

### 4.1. Results of the Virtual Object Automatic Scanning

The virtual object automatic scanning error (Figure 7a) was calculated by determining the distance between the axis plane that is automatically selected by the system and the axis plane where the actual target point is located. The average automatic scanning error was 1.008 mm with a deviation of 0.001 mm. The minimum and maximum errors were 0.978 mm and 1.039 mm, respectively. Due to the small deviation, the lower quartile, median, and upper quartile are represented by a straight line in the box plot. The box plot indicates two outliers with values of 0.978 mm and 1.039 mm, respectively. The target point error (Figure 7b) was determined using Avizo software.

To facilitate alignment in Unity, the anchor points of the two virtual objects were adjusted to the same position. Subsequently, the distance between the real target point and the virtual target point was used to obtain the target point error, which was found to be 1 mm with a deviation of 0.1 mm. The minimum and maximum errors were 0.495 mm and 1.21 mm, respectively, while the lower quartile, median, and upper quartile were 0.88 mm, 0.98 mm, and 1.12 mm, respectively. Stability tests (Figure 7c) were conducted in a total of 20 phantom trials. A script was written to record the center position of the scalp generated by the image target of the head every 1 s for 1 min. The stability was measured in 3 dimensions and normalization was performed afterward. The average stability and deviation were 0.076 mm and 0.052 mm, respectively.

### 4.2. Results of the Machine Learning to Predict and Recommend Surgical Target

The proposed system utilizes machine learning (specifically, the U-Net model) to predict and recommend surgical targets. This system was tested on 10 hydrocephalic patients, and the results indicated an average sensitivity of 93.85%, specificity of 95.73%, and accuracy of 99.93% in predicting the location of hydrocephalus.

The U-Net model generates a binary mask output, with ones indicating the ventricular region and zeros indicating other parts of the image. By comparing the output prediction to the labeled reference, true positive (TP), true negative (TN), false positive (FP), and false negative (FN) values can be computed. This allows for the calculation of sensitivity (TP/(TP + FN)), specificity (TN/(TN + FP)), and accuracy ((TP + TN)/(TP + FP + TN + FN)). Notably, the labeled reference of the ventricular region is available for experimental data, enabling the calculation of these indices in a similar manner.

The system enhances location prediction for hydrocephalus in terms of accuracy, sensitivity, and specificity. As shown in Table 3, the hydrocephalus prediction function presented in this paper can more accurately predict the location of hydrocephalus and provide surgeons with better surgical target recommendations, regardless of accuracy, sensitivity, and specificity.

### 4.3. Results of the Proposed System

The proposed approach exhibits fewer image navigation limitations and lower costs than optical navigation. A virtual object automatic scanning method is proposed to reduce calibration time in the preoperative stage, taking only 4 s. This represents an 87%, 96%, and 98% reduction in time compared to Konishi et al.’s ultrasonic scanning [33], Müller et al.’s reference registration method [5], and Tabrizi et al.’s projection registration method [2], respectively. Additionally, the proposed method achieves higher accuracy, with a range of 1 ± 0.1 mm, surpassing Ieiri et al.’s optical positioning navigation (18.8 ± 8.56 mm) [34], Deng et al.’s image positioning navigation (2.5 mm) [7], and Frantz et al.’s AR goggles surgical navigation system (1.41 mm) [9]. The proposed system leverages virtual object automatic scanning and image positioning for EVD surgical navigation, providing improved equipment cost, registration time, and accuracy.

Table 4 shows that the proposed image positioning method offers cost savings in comparison to the other three positioning methods, while also improving registration time and target accuracy. The registration time of 4 s is achieved through the virtual object automatic scanning method, while the accuracy of 1 ± 0.1 mm is obtained from the “3.1. Prosthesis experiment.” The proposed system provides superior target accuracy performance, primarily due to the virtual object automatic scanning method that offers accurate reference points, and all functions are performed within the system. This indicates that compared to other research methods, there are no external factors that may impact the accuracy of the target.

### 4.4. Results of the Hololens 2 Feasibility

Table 5 presents the accuracy results of five experiments conducted by five male experimenters aged 22 to 25, indicating that the impact of visual assistance with HoloLens 2 can differ significantly among users. Consequently, software feedback is essential for the navigation stick assistance method. However, as illustrated in Figure 8, the augmented reality performance of HoloLens 2 in tracking spatial images lacks stability, leading us to abandon the use of HoloLens 2 in the clinical feasibility test. For the clinical feasibility testing, a Microsoft Surface Pro 7 tablet was ultimately used.

## 5. Discussion

### 5.1. Comparing Augmented Reality Medical Guidance Techniques

#### 5.1.1. Image Positioning vs. Optical Positioning

Compared to the current optical navigation method, the proposed method in this paper offers significant advantages in terms of intuition, mobility, accessibility, and cost-effectiveness. Most current image-guided surgical navigation methods combine optical navigation with a navigation stick tracked by a cursor ball and display navigation information on an external screen.

Regarding intuition, our proposed method provides surgeons with intuitive spatial information through AR perspective display. In terms of mobility, the current optical navigation method requires a specific operating room, whereas our system can be used for surgical navigation in different fields with only a 10-min setup time. Furthermore, the proposed system is more accessible and cost-effective than the optical navigation method due to its lower equipment and practice costs. Augmented reality is an excellent solution for guiding surgery in areas with insufficient medical resources.

#### 5.1.2. Our Method vs. Other Method

Currently, several advanced augmented reality methods show promise for surgical navigation [36,37,38,39], but they still have limitations. Gavaghan et al. [36] used a projector to project liver blood vessels onto the liver surface, which has good accuracy but lacks deep spatial information and may not be suitable for guiding surgery. Kenngott et al. [37] proposed a method that provides three-dimensional anatomical information and has undergone clinical feasibility testing, but this method only offers viewing functions and lacks other auxiliary guidance. Heinrich et al. [38] and Hecht et al. [39] both provided visual aids for guided injections but lack feedback.

In comparison to the current augmented reality methods for medical guidance [36,37,38,39], the proposed method in this paper exhibits significant advantages in accuracy, provision of anatomical information, stereoscopic display, path navigation, visual feedback, and clinical suitability, as outlined in Table 6. As a result, the proposed method outperforms the current methods in all aspects.

#### 5.1.3. Tablets vs. Smart Glasses

The proposed system was implemented on a Microsoft Surface Pro 7 tablet and Microsoft HoloLens 2 smart glasses to compare their performance in terms of stability, flexibility, and information richness. The performance metrics are presented in Table 7.

In terms of stability, the Surface Pro 7 displays the head model and maintains the navigation stick’s stability well in a fixed position. On the other hand, the HoloLens 2 shows good stability for the head model in a fixed position, but its field of view changes with user movement, resulting in increased offset error. Additionally, the HoloLens 2 exhibits noticeable visual instabilities when tracking the navigation stick in motion.

Concerning flexibility, the Surface Pro 7 requires an additional stand that limits the viewing angle, while the HoloLens 2 has superior flexibility. Regarding comfort, the Surface Pro 7 is more comfortable as physicians do not need to wear the HoloLens 2, which can increase head weight, eye pressure, and visual interference.

Regarding information display richness, the HoloLens 2 can set windows in space to display DICOM information in a larger, clearer, and more persistent way. In contrast, the Surface Pro 7 can only display information on a single screen. Moreover, to avoid blocking the surgical guidance image, the DICOM image display must be canceled after selecting the target on the Surface Pro 7, preventing simultaneous display. Although multiple DICOM images can be superimposed and displayed on the head at the same time, the visual effect is not comfortable. Directly locking and displaying the target in an AR manner is a relatively simple visual effect after judging the target position.

Therefore, despite the HoloLens 2’s flexibility and complete information display advantages, guidance accuracy is the most critical factor, making the Surface Pro 7 the ideal platform for implementation.

### 5.2. The Impact of Viewing Angle Changes on the Coordinates

To discuss the intrinsic error of Vuforia, the influence of the head model and the tip position of the navigation stick on the coordinate position at different visual angles was tested. Figure 9 presents the results of testing commonly used navigation stick angles (60~140°) and head model recognition, graphing their influence on coordinates under reasonable usage angles (60~120°). The X coordinate of the navigation bar is found to be significantly affected by the viewing angle, but within the most frequently used range of angles (80~100°), the error is only ±1 mm, and there is no significant effect on the Y and Z coordinates, with most errors outside the outliers within ±1 mm. Additionally, to examine the impact of changes in the viewing angle of the head model identification map, the system was tested in the range of 60–120 degrees in 10-degree increments, as both the head model identification map and the camera’s viewing angle are fixed values.

### 5.3. Sterile Environment for Surgery

To ensure suitability for clinical applications, the proposed system must be able to function in a sterile environment. As such, a layer of the surgical cell membrane is covered on the identification map, and the tablet is wrapped in a plastic sleeve, allowing it to remain operational without compromising sterility.

### 5.4. Clinical Significance and Limitation

In summary, previously proposed methods have not provided a comprehensive solution for accurately guiding surgical targets, scalpel entry points, and scalpel orientation in brain surgery. The proposed approach aims to address these shortcomings. To ensure ease of portability, a tablet PC is used as the primary AR device in the proposed system. The DICOM data processing takes approximately one hour to complete the system update. Surgeons can use the proposed system before and during surgery for real-time guidance on surgical targets, entry points, and scalpel paths. In terms of precision, the proposed system has an average spatial error of 1 ± 0.1 mm, which is a significant improvement over many previous methods. The system achieves improved recognition accuracy, sensitivity, and specificity, with values of 99.93%, 93.85%, and 95.73%, respectively, marking a significant improvement over previous studies. Smart glasses are not recommended for the proposed AR system due to their potential to introduce significant errors, as accuracy and stability are important considerations.

## 6. Conclusions

This study combined virtual object automatic scanning with deep learning and augmented reality to improve surgeons’ surgical procedures. U-Net was utilized for deep learning to predict the location of hydrocephalus, reducing pre-operative time requirements and increasing surgical precision. Augmented reality overlays virtual images directly on the patient’s head, allowing for intuitive guidance in locating surgical target points and trajectory guidance for improved accuracy in EVD surgery. The proposed system also employed color coding for angle correction at the surgical entry point, allowing for more intuitive and accurate operations. The developed EVD surgical navigation system using virtual object automatic scanning and augmented reality shows improved accuracy, registration time, and surgical costs. Future work will focus on exploring the use of smart glasses for collaborative operations and conducting clinical trials for intraoperative navigation to enhance the clinical utility of the proposed system.

## Figures and Tables

**Figure 1 bioengineering-10-00617-f001:**
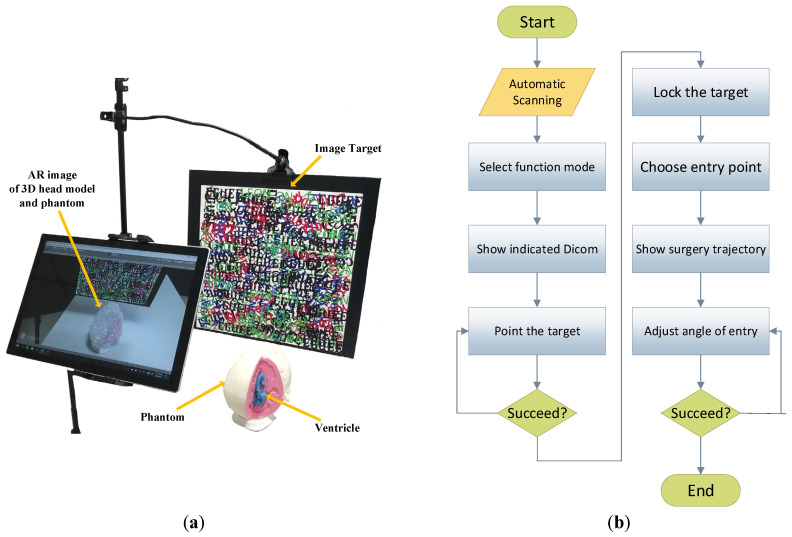
Surgical navigation overview and process flow chart. (**a**) Hardware overview of surgical navigation, and (**b**) flow chart of surgical navigation.

**Figure 2 bioengineering-10-00617-f002:**
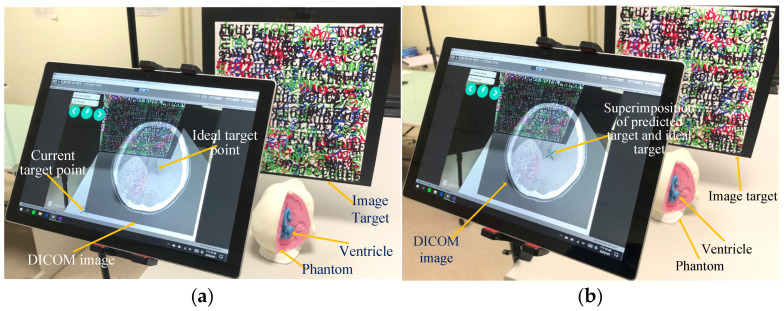
Screen display of target selection. (**a**) Before target selection, and (**b**) after target selection.

**Figure 3 bioengineering-10-00617-f003:**
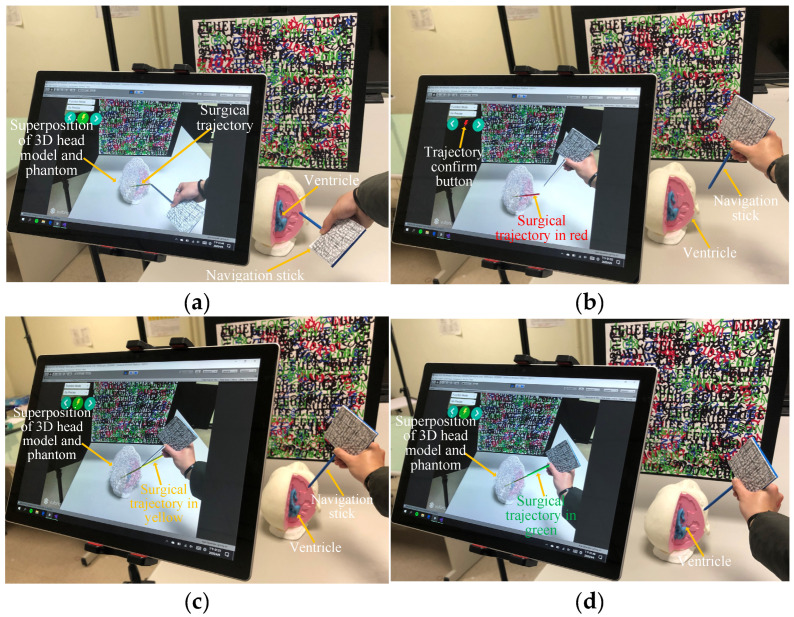
Calibration of surgical trajectory and entry point. (**a**) Surgery trajectory, (**b**) deviation from the entry point, (**c**) approach to the trajectory, and (**d**) calibrated trajectory.

**Figure 4 bioengineering-10-00617-f004:**
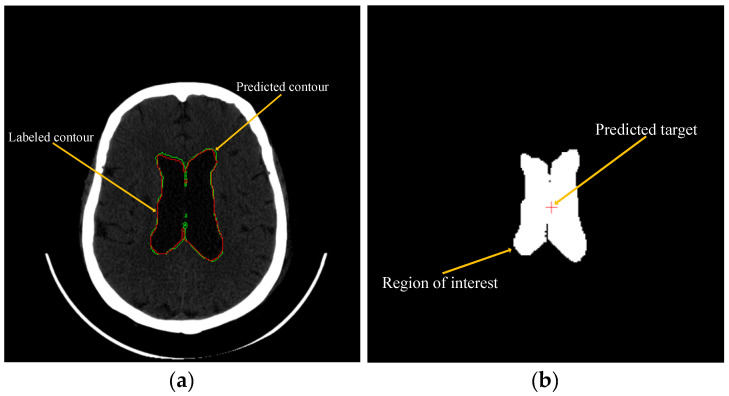
Comparison of contour and indication of the predicted target. (**a**) Contour comparison, and (**b**) predicted target indication.

**Figure 5 bioengineering-10-00617-f005:**
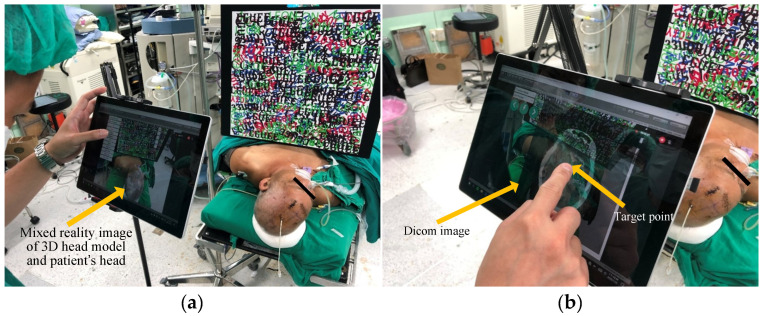
Clinical feasibility test. (**a**) Superimposition of the 3D model onto the patient’s head, (**b**) selection of target point, (**c**) calibration of trajectory angle, and (**d**) completion of calibration.

**Figure 6 bioengineering-10-00617-f006:**
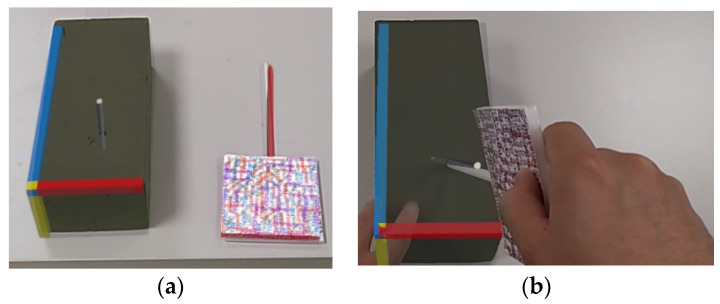
Accuracy experiment of AR overlay using HoloLens 2. (**a**) Superimposing the real sponge brick and the virtual model, and (**b**) aligning the navigation stick with the path of the virtual guide.

**Figure 7 bioengineering-10-00617-f007:**
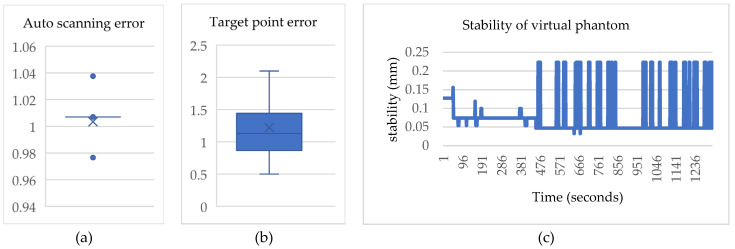
Boxplot analysis of automatic scanning and target point error. (**a**) Automatic scanning error, (**b**) target point error, and (**c**) line graph illustrating stability test on a virtual phantom.

**Figure 8 bioengineering-10-00617-f008:**
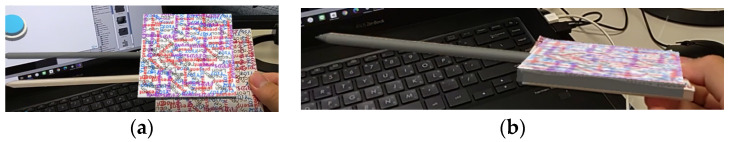
Spatial image tracking using augmented reality representation on HoloLens 2. (**a**) Imprecise, and (**b**) Precise.

**Figure 9 bioengineering-10-00617-f009:**
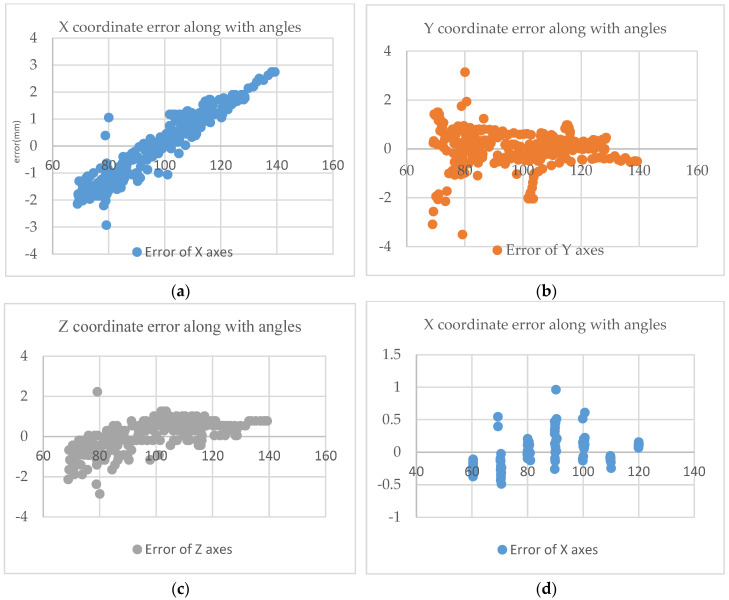
Effect of Viewing Angle Changes on Coordinate Accuracy. (**a**) X-coordinate error at various navigation stick angles, (**b**) Y-coordinate error at various navigation stick angles, (**c**) Z-coordinate error at various navigation stick angles, (**d**) X-coordinate error at various phantom angles, (**e**) Y-coordinate error at various phantom angles, and (**f**) Z-coordinate error at various phantom angles.

**Table 1 bioengineering-10-00617-t001:** Notations.

Notation	Definition
$Dis_{x}$/$Dis_{y}$/$Dis_{z}$	Distance on the *x*/*y*/*z* axis of the DICOM object
$E_{p}$	The edge point of the scalpel
$l_{dicomX}^{2D}$/$l_{dicomY}^{2D}$	The length/width of the DICOM image
$l_{X}^{3D}$/$l_{Y}^{3D}$/$l_{Z}^{3D}$	The length of the head along the *x*/*y*/*z* axes
$N_{target}$	The number of the specific DICOM slice with the ideal target
$N_{total}$	The total amount of DICOM slices
$Num_{x}$/$Num_{y}$/$Num_{z}$	The number of DICOM slices on the *x*/*y*/*z* axis
$Pos_{target}^{2D}(X_{target}^{2D},Y_{target}^{2D})$	The ideal 2D target position
$Pos_{target}^{3D}(x,y,z)$	The ideal 3D target position
$Pos_{0}^{3D}(0,0,0)$	The origin point
$RP_{x}{}^{L}$/$RP_{x}{}^{R}$	The reference point on the left/right of the *x*-axis
$T_{x}$/$T_{y}$/$T_{z}$	The thickness on the *x*/*y*/*z* axis of DICOM slices
TrueX/TrueY/TrueZ	The *x*/*y*/*z*-axis displayed the DICOM slice

**Table 2 bioengineering-10-00617-t002:** Definition of common abbreviations.

Abbreviation	Meaning
AR/VR	Augmented reality/Virtual reality
AR devices	Augmented reality devices, which include mobile devices and head-mounted displays, that are capable of running augmented reality software
CNN	Convolutional neural network. A type of artificial neural network that belongs to the family of deep learning algorithms
EVD	Extra-ventricular drainage
DICOM	Digital Imaging and Communications in Medicine
DICOM files	Files that have been saved in the Digital Imaging and Communications in Medicine (DICOM) format, typically obtained from computed tomography (CT) or magnetic resonance imaging (MRI) procedures.
DICOM data	Data that has been formatted in accordance with the Digital Imaging and Communications in Medicine (DICOM) standard
DICOM image	A 2D image that has been saved in the Digital Imaging and Communications in Medicine (DICOM) format and can be displayed on a screen or monitor
DICOM slice	A single two-dimensional image of a patient’s anatomy that has been acquired using imaging modalities such as X-ray, CT scan, MRI, or ultrasound.
HMD	Head-mounted device
Virtual image	Computer-generated image that cannot exist in the physical world and can only be viewed on a display monitor
Virtual object	A digital object that lacks a physical presence and can only be created and manipulated within software
U-Net	U-Net is a convolutional neural network that was originally developed at the Computer Science Department of the University of Freiburg for biomedical image segmentation. The original U-Net model, which consists of an encoder and decoder, is widely used for tasks such as medical image segmentation, as presented in this paper.

**Table 3 bioengineering-10-00617-t003:** Comparison of accuracy, sensitivity, and specificity.

Method	Accuracy (%)	Sensitivity (%)	Specificity (%)
Proposed	99.93	93.85	95.73
Surgeon	85–97	N/A	N/A
[30]	N/A	90.0	85.0
[31]	91.80	91.84	94.77
[32]	N/A	80.0	90.0
[2]	94.0	93.6	94.4
[10]	93.0	N/A	N/A

**Table 4 bioengineering-10-00617-t004:** Comparison of registration time, average error, and standard deviation.

Method	Registration Time	Average Error (mm)	Standard Deviation (mm)
Proposed	4 s	1	0.1
[7]	30 s	1.96	0.87
[8]	99 s	2.5	N/A
[35]	N/A	2.1	N/A
[9]	228 s	1.2	0.54
[34]	N/A	3	N/A
[13]	N/A	2	N/A
[36]	N/A	2	N/A
[33]	N/A	1.9	0.45
[4]	N/A	2.8	2.7
[5]	300 s	7.9	N/A

**Table 5 bioengineering-10-00617-t005:** Accuracy results of HoloLens 2 feasibility test.

Experimenter	A	B	C	D	E	Average
Error (mm)	5	3	7	2	13	6

**Table 6 bioengineering-10-00617-t006:** Comparison of properties between our method and other methods for using augmented reality in medical guidance.

Properties	[36]	[37]	[38]	[39]	Proposed
Device	Handheld projector	iPad	HoloLens 1	Mobile phone	Surface Pro 7
Accuracy	projection error: 1.3 ± 0.9 mm	Prosthesis experiment: 2.8 ± 2.7 mm	Average insertion angle difference: 6.35 degree	Prosthesis experiment: 2.7 ± 2.6 mm	Prosthesis experiment: 1.0 ± 0.1 mm
Anatomy information providing	V	V			V
3D display providing		V	V	V	V
Path navigation providing			V	V	V
Visual feedback providing					V
Clinical feasibility	V	V		V	V

**Table 7 bioengineering-10-00617-t007:** Feature comparison of Surface Pro 7 and HoloLens 2.

Features	Surface Pro 7	Hololens 2
Stability	Better	
Flexibility		Better
Comfort	Better	
Information richness		Better

## Data Availability

The statistical data presented in this study are available in Table 3 and Table 4. The datasets used and/or analyzed during the current study are available from the corresponding author upon request. These data are not publicly available due to privacy and ethical reasons.

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
