# Peer review of "Augmented Reality Surgical Navigation System Integrated with Deep Learning"

_bioengineering, 2023, doi:10.3390/bioengineering10050617_

Round 1

Reviewer 1 Report

The aim of this paper is to investigate the Augmented Reality Surgical Navigation System Integrated with Deep Learning. In my opinion that this paper has no novelty. However, mycomments as follows: 

1. The abstract requires improvement in terms of structure and content. Providing more detailed information on the proposed model and results is recommended. 

2. The introduction should also provide a more comprehensive explanation of deep learning methods. Some recommended sources that can be used for reference are: https://doi.org/10.1016/j.inffus.2022.12.010 and https://doi.org/10.1007/s11571-022-09897-w. 

3. To further improve the paper, the authors should add a "Related Works" section that provides a comprehensive literature review. A tabular summary of the paper's review should be included to give readers a better understanding of the research conducted in this field. This table should include information about the works, dataset, preprocessing, main methods and performance in percentage. Some articles should also be presented in the form of text.

4. Please add more details for your deep learning model like hyper-parameters,  learning curves and loss curves. 

5. Simulating some basic methods and comparing the results with the proposed model would also be helpful.

6. The research question(s) needs to be more clearly defined, and the initial hypothesis clarified. 

7. The Performance Metrics should be summarized in a table, and in the Discussion section, the authors should critically discuss the work/results against the hypothesis. 

8. Key findings should be identified, and the novelty and contribution of the work justified.

9. A summary of all relevant parameters, with clear explanations, should be provided to aid readers. 

10. The clinical significance of the findings should be highlighted, and a section on the study's limitations should be added.

11. In the Conclusion section, the authors should elaborate more on future work, and a table should be included comparing the proposed method with related works.

12. While the overall English language is acceptable, some errors need to be corrected.

Author Response

Dear reviewer,

Please see the attached file. 
Thank you for your time and your help.

Best regards,

Shin-Yan Chiou
Professor,
Department of Electrical Engineering,
Chang Gung University

Reviewer 2 Report

      This paper considers the development of an augmented reality (AR) surgical navigation system with advantages of low cost, high stability and high accuracy. The proposed system can show the connection between the surgical target and the surgical entry point on an AR device and provide intuitive guidance for the surgical target point, entry point and trajectory. The results of clinical trials on extra-ventricular drainage (EVD) confirm the feasibility and accuracy of the proposed system. The proposed system is original and may have its own merits in clinical practice. The following points need to be addressed before the paper can be accepted for publication.

1. The paper should emphasize and elaborate on the motivations and major contributions of the paper in the end of Section 1.

2.   The authors may need to provide the full name for DICOM (line 120, page 3) in Section 2.2.

3.    In Section 2.2, can the authors provide a clearer explanation on the principle of trigger function and how the six reference points are used to locate the DICOM plane (lines 116 and 123, page 3) ?

4.  In Section 2.3, how many patients does each group contain (line 127, page 4)? can the authors provide more information on the structure of the UNET model they have trained to predict and recommend surgical entry points (line 136, page 4)? What is the overall accuracy of the trained model?

5.  In Table 2 (line 221, page 7), how are the sensitivity, specificity and accuracy calculated based on experimental data?

6.  Table 5 (line 362, page 13)  needs better explanations. For example, what do letters V and X represent respectively? 

Author Response

(The authors gave the same response as above.)

Reviewer 3 Report

This manuscript described a AR based navigation system with additional improvements. Reads  quite a few interesting  information. But the presentation of test of this new system is insufficient. Crumbing information into two sections:  "3. Experiments and Results experimental study and 4. Discussion" without proper organization is not acceptable. The introduction of the proposed system is clear but the tests involved, including clinical feasibility study lacks of clear definitions , and therefore  relevant results are questionable.  For example,  it is concluded that "the system achieves recognition accuracy, sensitivity and specificity respectively of 99.93%, 93.85% and 95.73%, marking a significant improvement on 34
previous studies." but readers does not even know how many data were involved in these studies.  To be frankly,  I see a collection of data/numbers/photos, but no sound scientific studies were reported in manuscript. Please rewrite section and 3 and 4, separate them into Experimental Studies and Tests, Results, and Discussion, and provide sufficient information in each tests/experiments.

I am glad to review your reorganized version.

Author Response

(The authors gave the same response as above.)

Round 2

Reviewer 1 Report

Thanks for your revision, I strongly recommended the new version of the paper can be accepted for publication. 

Author Response

Please

Reviewer 2 Report

All issues have been addressed.  I have no other concerns and recommend the acceptance of the paper.

Reviewer 3 Report

This revised version presented with a clear and improved structure, thus allow  further review in detail. Thanks authors' effort. However, the restructure need to be completed entirely. The study itself has the merits to be published but the format and structure  needs to reach the minimal requirements of a journal article.

The format should follow a journal article, not a thesis/dissertation/grant proposal.

All references need to be renumbered.

A list of abbreviation helps.

All captions of tables and figures should be self-content.

Please be concise. At least 10% can be either deleted or rewritten. Please start from remove/rewrite "we" sentences, ...

In addition, I offer the following comments, questions and suggestions for your consideration:

L24-25: This statement is confusing. Can you distinguish among (1) optical (or one of conventional) navigation system, (2) optical navigation system with AR ( if such system existed?), and (3) AR navigation system?

L30: Which AR device? Hololens? Smart Phone? or others?

L32: "Confirming feasibility"? Sounds awkward.

L33-34: High accuracy of what? What this method for?

L34: Please specify which neural networks algorithm.

L38: Does this new system is AR based or MR based? Why both keywords are listed here?

L42: Joda et al. [25] is a review article. Did they, has someone else already,  suggested AR is useful in complex procedures? Besides, It is important to apply knowledge based review/introduction.

L43: "This study"? Which one? Applied all three type of navigation in one study?

L48: why new paragraph?

L54-55: "subjecting ... to the TLD method"?

L80: This sentence is a wordy. Please make it simple.

L82:  There is no " more particularly". in particularly?

L87: " Note that, ..." ?  Also UNET needs reference(s).

Please allow me speak frankly, the Introductionsection is poorly written. Firstly, although every specific reference is carefully summarized, the soul of the introduction is lost: what is the clinical problem and /or surgical procedure to deal with? What technology is/are available? why this study/new approach is proposed in this study? etc. Secondly, the key component is missing:  what is goal/objective/aim of this proposed study? what problem this study intends to solve? L105-L117 need to define the objective of this study,  and perhaps, what approach intends to adapt. The methododology and results follows.

It is usually better to select information to build up the knowledge base and  to answer the aforementioned questions, while put the rest of literature review to the discussion section.

L 118 "2 Related works" should be combined into Introduction.

L134: Please include the vendor/manufacturers information for each commercial products.

L137: Figure 1 should follow and be placed to near this sentence.

L143: please clarify "virtual image" . May need a better , more specific term.

L162: What is ML? please clarify.

L163: DICOM, here,  is only a file format. Please clarify. How many patients in these 10 groups? in each group?

L143-189: if only these descriptions may follow the Figure1 ...

L206 and others:  Please numbering each formula properly and consistently.

L231: Why does calibration after the manually select the entry point? The details suggests a confirmation from the surgeon. This is not the step of calibration, at least , from an EE engineer perspective.

L246. Why not delete the last sentence?

L251-270: these are results of Prosthesis experiment. and belongs to the Results section. Please describe  the experiment itself, such as, the set up, materials/ prosthesis, experimental procedure, parameters to be measured, data collected,  and analysis performed. Where does figure 5 come from?

L271-280: This is better. Figure 6 should follow this paragraph. Also, please add further details. please see my comments to the prosthesis experiment.

L279-280: This sentence is a part of the results of this test.

L281-296:  Loos there is other tests than those two tests . Please add it to the method then, logically before the prosthesis experiment. Please describe how it is performed, what data used and what parameters were  extracted, and how the accuracy was evaluated, etc..

L292: does this 10 patients are subjects of  clinical feasibility test? If so, please report the results clearly, with subtitle, for instance.

Table 2 will loos better without N/A. Also, Registration time (s).

Table 2 and 3 , which test?

The results section need to be further organized and presented in a way to allow one to draw conclusion(s) logically.

L337: What is smart glasses? Is it in your system? If so, why not describe it in the method?

L337-351: Is this an small, additional test? please clarify.

Table 4 is perhaps the most important results of this system, which against all previous study with a  smallest error of 2 mm. This is suggests the new proposed system does not fulfill the clinical requirement yet.

Line 384 "7 Comparison" studies should be listed in the method , and their results should listed in the results.  The restructure has not completed. Please do so.

Overall impression, if the original manuscript is a thesis /dissertation, it need to be restructured and reformed to be a publishable article. Please do so!

Looks there is a long way to go based upon these two rounds of review. Please DO rewrite and reorganize following a typical journal article format, to the minimum. Please resubmit when the manuscript reaches the BASIC requirements.

Round 3

Reviewer 3 Report

This is the third round of review of this interesting study. This reorganized version is much, much better, and it should be the version ready to be submitted for publication. Congratulation! You made it!

There is one issue I want to make it clear to all authors before we move forward:

(1) Never, ever "(It is) assumed that readers familiar with this specialized field (would understand the difference between these two systems)." Our readers are not necessarily an "experts in this field." but you are. The Journal article you prepare intends to reach larger audience. This is a fundamental difference between a thesis/dissertation (for your committee) and journal article ( for general readers). Please appreciate it. On the other hand, providing too much basic knowledge would make the manuscript lengthy. It is an art we must master to strike a delicate balance.

Reading through this new version, I offer the following for your consideration:

(2) This revised version (V3) still does not provide a clear description on the the objective/aim of this study: The end of introduction reads like a summary of this entire study. It is perhaps clearer if the last two paragraphs (L123-134) are removed/deleted. Of course, you can provide a much better description of the objective/aims of this study, from proposed methodology to its clinical validation (even just a case report), and anything in-between.

(3)The introduction is too long for a journal article. I recommend keep the major points while move detailed literature review to Discussion. Please refer to my comments to your previous version.

(4) The writing can be more concise.

In addition, some details as well:

(4) L34: "a high accuracy of 1±0.1mm " ? If you use CT scan with a spatial resolution of 0.5 mm,  the best accuracy you may reach is 0.5 mm. 

(5) L42: VR? Does this manuscript involve VR? Please focus on the topic.

(6) L43: "AR/VR/AI to be combined with medicine"?  How about "applied"? 

(7) L44 and L64: citation error/conflict.

(8)L76: VT?

(9) L77: how about "the use of AR" instead of "AR use"?

(10)L87-88: " surgical medical guidance"? typo?

(11) L90: ? this sentence need to be rewritten.

(12) L123-132: These two paragraphes does not belong to here. Delete or move.

(13) L17: Surface Pro 7 (Microsoft Co.,  Redmond, WA, USA)

(14) Fig1 presented MR. Why MR was not introduced in Introduction? Please be consistent.

(15) Table 2: VR? any thing to do with Vr in the study?

(16) Table 2:  A list of abbreviations is usually placed either before the introduction or after the conclusion. There is no need to provide a table. Please discuss with the editor about this. However, I consider table 2 is a list of definitions. DICOM image need further clarification: is it a 2D image or cross section image of a CT scan, either orthogonal or oblique? or is it a 2D display of 3D object in a 3D space( virtually)? 

(17) Table 2:  U-NET, which version is used in this study? 

(18) L156: Please provide vendor information for Avizo. Please refer to (13).

(19) L166-167: What is "DICOM formatted displays"? this is not listed in Table 2, is not it?

(20) The word "scan" is confusing with " CT scans". 

(21) L172-173: What do you mean " MR is generated"?

(22) L175-176: How does " the system selected the connection areas"?

(23) L178-186: following Table 2, DICOM files are either CT or MRI scan data sets. How many sets of DICOM files in each group? and How many patients/subjects were involved, considering there may be multiple CT/MRI data sets for a particular patient?

(24) L184-186: How does DICOM files become "Normalized data"  ?  How does "normalization" were performed?

(25) L188: How does the superimposition of virtual head to real head be performed? What tool to use?

(26) DICOM slice is commonly used term. Which term in Table 2 is equivalent to DICOM slice?

(27) L245: How do you determine the " actual axis planes" from 10 patients?

(28) L248: Please vendor information for 3DSMax. and please using the format as shown in (13).

(29) L255: so does to Matlab

(30) L288: ?
